# Adaptation optimizes sensory encoding for future stimuli

**Jiang Mao** [1], **Constantin A. Rothkopf**[2], **Alan A. Stocker** [1]*

**1** Department of Psychology, University of Pennsylvania, Philadelphia, Pennsylvania, United States of America, **2** Institute of Psychology, Technical University Darmstadt, Darmstadt, Germany

* astocker@psych.upenn.edu

**Data availability statement:** Raw data and code for modeling and analyzing the data are freely available at the following online repository: https://github.com/cpc-lab-stocker/adapt-discr-efficient-code.

**Funding:** JM was in part supported by the National Science Foundation and DoD OUSD (R&E) under cooperative agreement PHY-2229929 (The NSF AI Institute for Artificial and Natural Intelligence ARNI) with AAS. The funders had no role in study design, data collection and analysis, decision to publish, or preparation of the manuscript

## Abstract

Sensory neurons continually adapt their response characteristics according to recent stimulus history. However, it is unclear how such a reactive process can benefit the organism. Here, we test the hypothesis that adaptation actually acts proactively in the sense that it optimally adjusts sensory encoding for future stimuli. We first quantified human subjects' ability to discriminate visual orientation under different adaptation conditions. Using an information theoretic analysis, we found that adaptation leads to a reallocation of coding resources such that encoding accuracy peaks at the mean orientation of the adaptor while total coding capacity remains constant. We then asked whether this characteristic change in encoding accuracy is predicted by the temporal statistics of natural visual input. Analyzing the retinal input of freely behaving human subjects showed that the distribution of local visual orientations in the retinal input stream indeed peaks at the mean orientation of the preceding input history (i.e., the adaptor). We further tested our hypothesis by analyzing the internal sensory representations of a recurrent neural network trained to predict the next frame of natural scene videos (PredNet). Simulating our human adaptation experiment with PredNet, we found that the network exhibited the same change in encoding accuracy as observed in human subjects. Taken together, our results suggest that adaptation-induced changes in encoding accuracy prepare the visual system for future stimuli.

## Author summary

Prolonged exposure to a fixed stimulus causes sensory neurons to adapt. In this study, we uncover some of the functional benefits of adaptation for the visual system. We first quantified how adaptation changes the sensory representation of a stimulus feature (here, local stimulus orientations) using psychophysical measurements. We found that adaptation improves the sensory representation of stimulus orientations that are similar to the adaptor orientation, while it weakens the representation of dissimilar orientations. By analyzing the retinal image statistics of freely behaving human subjects, we then show

**Competing interests:** The authors have declared that no competing interests exist.

that these enhanced representations are tailored to the immediate future retinal input because stimulus orientations are more likely to be similar to those in the past. Finally, we show that an artificial neural network trained to predict the next frame in naturalistic videos exhibits changes in sensory representation that are very similar to the ones measured in human subjects. Together, our results indicate that adaptation improves sensory representations in a way that benefits the visual system in processing future sensory input.

## Introduction

Biological information processing systems continually adapt their sensory representations to statistical changes in their sensory environment. This is well documented by the various neural response changes that occur after prolonged exposure to, for example, a fixed visual orientation [1–3] or motion direction stimulus [4,5], but also the corresponding perceptual changes [6–8]. Many popular visual illusions, such as the motion aftereffect [9], represent particularly salient examples of how adaptation affects perception. Adaptation manifests itself at every stage of information processing, affecting every neuron involved in the representation and processing of sensory information, and the effects accumulate and interact along the representational hierarchy in the brain (e.g., [5,10]). Its ubiquitous nature suggests that adaptation provides fundamental and important benefits [11–13].

Adaptation has been thought of as a mechanism that adjusts neural representations in order to maximize the amount of information encoded about sensory input [14–18]. This efficient coding hypothesis has been empirically validated at the level of single neurons in simple systems where input-output relations can be readily controlled and measured (e.g., the motion sensitive neurons of the blowfly [19,20]). In more complex neural systems such as the primate brain, however, perceptual variables are typically encoded in a distributed fashion over entire neural populations. In this case, the efficient coding hypothesis is more difficult to test because it requires sufficiently comprehensive measurements of the neural code to faithfully calculate the information content across the entire neural population. Using Fisher information [21] as a measure of encoding precision can help to resolve these difficulties. Fisher information is an approximation of the computationally more demanding measure of mutual information [22], and can be interpreted as the amount of coding resources allocated to represent a particular stimulus value [23]. It can be directly computed from and related to neurophysiological parameters [24–28]. More importantly, Fisher information also provides a bound on perceptual discriminability [24,29,30]. As a result, adaptation-induced changes in sensory encoding precision can be directly quantified with psychophysical discrimination experiments.

Previous studies have found that discrimination thresholds decrease for stimulus values close to the adaptor but increase for values different from the adaptor [6,7,31,32]. Some studies reported that discrimination thresholds also decrease for stimulus values that are opposite of the adaptor (i.e., for orientation, orthogonal to the adaptor) [2,7], but this result is not conclusive [33,34]. However, because these previous psychophysical studies did not measure the effects of adaptation over the entire stimulus range, the currently available discrimination data are not sufficient to fully characterize adaptation-induced changes in sensory encoding.

Testing the efficient coding hypothesis of adaptation also requires knowledge of the contextual stimulus distribution for which the representation is optimized. Previous studies have shown that under (quasi-)stationary contexts sensory encoding is qualitatively well matched

to the overall, longterm statistics of the observer's natural environment. For example, the distribution of local orientations in natural visual scenes shows strong peaks at cardinal orientations [35] that are well aligned with the reported higher orientation discriminability of human observers at cardinal orientations [27,36,37]. However, these input distributions are more difficult to define and measure at the relative short timescales relevant for adaptation. Particularly in vision, the observer's active control of gaze position can substantially affect the shape of the input distributions at the level of the retina [38–40]. Efficient coding seems also somewhat at odds with the notion that adaptation is driven by stimulus history: to be beneficial, sensory representations should be optimized for future rather than past sensory input. Of course, it is well established that due to its continuous nature, the recent state of our environment is a good predictor of its future [2,41–43]. Again, this does not necessarily translate to the input distributions at the level of the retina because of the observer's active control and selection of sensory input via eye-movements [44–48].

With our study, we validated the efficient coding hypothesis of adaptation for visual orientation perception. We psychophysically characterized changes in visual orientation discriminability across the entire orientation range induced by prolonged exposures to different adaptor stimuli with different orientations. We found that adaptation increases discriminability at the adaptor orientation and also (mildly) at orientations orthogonal to the adaptor, but otherwise decreases discriminability away from the adaptor orientation. Adaptation is therefore best described as a reallocation rather than a change in the overall amount of coding resource, which can be expressed with a single, isomorphic adaptation kernel for the adaptation conditions in our experiments. By analyzing the retinal input of freely behaving human observers in natural outdoor environments, we show that this kernel is qualitatively optimized for the natural orientation distributions of future stimuli at short timescales under qualitatively comparable adaptation conditions. Finally, we use an artificial recurrent neural network, designed and trained to predict the next frame of naturalistic videos [49,50], to demonstrate that these adaptation kernel naturally emerge in this neural network when being presented with the same adaptation stimulus sequence used in our human psychophysical experiment. Taken together, our results suggest that adaptation helps to maintain an efficient representation of future stimuli given the shortterm, temporal context of sensory signals experienced under natural conditions.

Some of the conceptual ideas of the presented work are inspired by earlier proposals [51, 52]. Preliminary results have been presented at the Annual Meetings of the Vision Science Society in May 2020 and 2023, respectively [53,54].

## Results

The total encoding capacity of a biological sensory system is limited, i.e., the precision with which sensory information can be represented is finite. We hypothesize that adaptation temporarily reallocates some of the encoding capacity in order to optimize encoding for statistically more likely future sensory input. In the following, we first formalize the reallocation process before we empirically test the hypothesis.

### Reallocation model

We use Fisher information $J(\theta)$ to quantify encoding precision. Fisher information provides a lower bound on discrimination threshold $D(\theta)$ [24,29] given as

$$\sqrt{J(\theta)} \propto 1/D(\theta), \tag{1}$$

which allows us to directly measure encoding precision with appropriate psychophysical discrimination experiments. Furthermore, the efficient coding hypothesis links Fisher information to the stimulus (i.e., prior) distribution $p(\theta)$. For example, for efficient representations that aim to maximize mutual information between stimulus and encoded values, this dependency is of the form [22,23,37]

$$\sqrt{J(\theta)} \propto p(\theta). \tag{2}$$

Note that for other efficiency criteria the constraint (Eq 2) differs yet generally remains a power-law function for a large family of objective functions [55,56]. Thus with the above assumptions, Fisher information provides a link between encoding precision and psychophysical measures of discriminability (Eq 1) as well as the stimulus distribution for which encoding is optimized for (Eq 2).

Now, let us consider the sensory representation of local stimulus orientation in the visual system (Fig 1). Under (quasi-)stationary conditions (i.e., at long timescales), we assume higher encoding precision for cardinal orientations. This aligns with efficient coding given that the overall distribution of local orientations in natural scenes typically shows prominent peaks at cardinal orientations (see also [35,55,57]). Depending on the specific natural scene database, these peaks can be more or less symmetric with regard to vertical and horizontal. For reasons of simplicity, we assume a symmetric model of the stationary encoding precision (see Methods).

We can define a new space $\tilde{\theta} = F(\theta)$ for which the distribution of Fisher information is uniform and encoding is homogeneous under stationary conditions [23,28]. We refer to this new space as "sensory space". Given the structural homogeneity of the cortical sheet in visual cortices, we consider this space a one dimensional proxy for the neural representation of visual orientation. $F(\theta)$ can be thought of as the projection of stimulus information onto this internal, sensory representation. Because adaptation is intrinsically a neural process, reallocation is best described at the level of this sensory space.

After prolonged exposure to an adaptor stimulus with a single orientation, we assume that the encoding capacity is temporarily redistributed depending on the shortterm stimulus distribution during adaptation (Fig 1b). In our experiment, this distribution is narrowly centered around the adaptor orientation (22.5 or 45 deg), and thus is approximately identical in sensory space for the two adaptor orientations considered here. As a result, the model assumes that the reallocation process in sensory space can be described with an isomorphic adaptation kernel, i.e., the reallocation of sensory coding resources follows a fixed pattern relative to the orientation of the adaptor stimulus that is independent of the adaptor orientation. If reallocation is aimed at maintaining an efficient representation, the kernel should reflect the changes in stimulus distribution for which adaptation is meant to optimize sensory encoding (Eq (2)). A final assumption of the reallocation model is that the total encoding capacity (i.e., the total Fisher information) does not change with adaptation, meaning the average sensory noise as expressed by the width of the sensory measurement distribution in sensory space remains constant.

With the reallocation model, we can directly extract Fisher information and its changes from psychophysical discriminability measurements under different adaptation conditions. Likewise, we can predict the psychometric functions of the discrimination experiment that correspond to a given adaptation kernel (see Methods for details). We use the latter to extract the most probable adaptation kernels from fits to the psychophysical discrimination data.

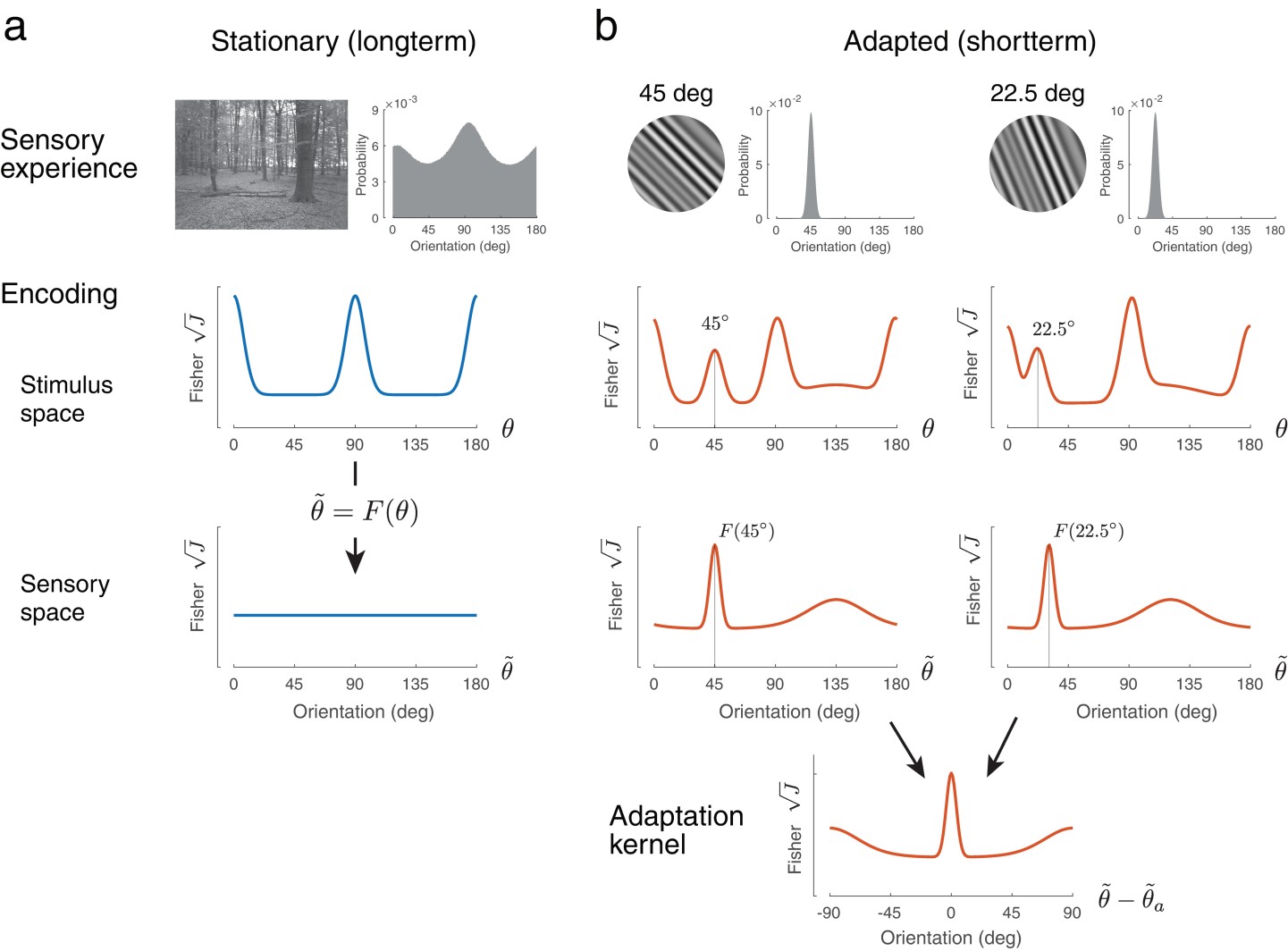

**Fig 1. Reallocation model.** (a) Under stationary conditions, sensory encoding capacity (in units of square-root of Fisher information) is allocated according to the longterm stimulus distribution (Eq (2)). For visual orientation $\theta$, this distribution exhibits characteristic peaks at the cardinal orientations. Shown is one sample frame and the overall orientation distribution of the retinal video sequences we analyzed in our study. We define a sensory space $\tilde{\theta} = F(\theta)$ such that encoding precision in that space is uniform. (b) After adaptation (shown are the two oblique adaptor stimuli used in our psychophysical experiments), encoding capacity is reallocated depending on the shortterm stimulus distribution. Adaption to a single, narrow orientation-band adaptor stimulus, results in a narrow shortterm distribution. The model assumes that in this case, encoding capacity is reallocated according to an isomorphic adaptation kernel in sensory space, i.e., the kernel is identical in shape for different adaptors but centered at the respective adaptor orientation. Both, stationary encoding capacity and the adaptation kernel can be obtained from joint, parametric fits to the psychometric functions obtained from the psychophysical discrimination experiments. Total encoding capacity is assumed to remain constant across all conditions. Note that the widths of the shortterm stimulus distributions are exaggerated for illustration purposes.

## Psychophysical experiment

Five human subjects performed a two alternative forced-choice (2AFC) orientation discrimination experiment under different adaptation conditions (Fig 2). At the beginning of each block, subjects were presented with an adaptor stimulus for one minute. After that, subjects performed 192 trials of the 2AFC task, reporting which one of two orientated grating stimuli was more clockwise/counter-clockwise. Every trial started with a 5s period of top-up

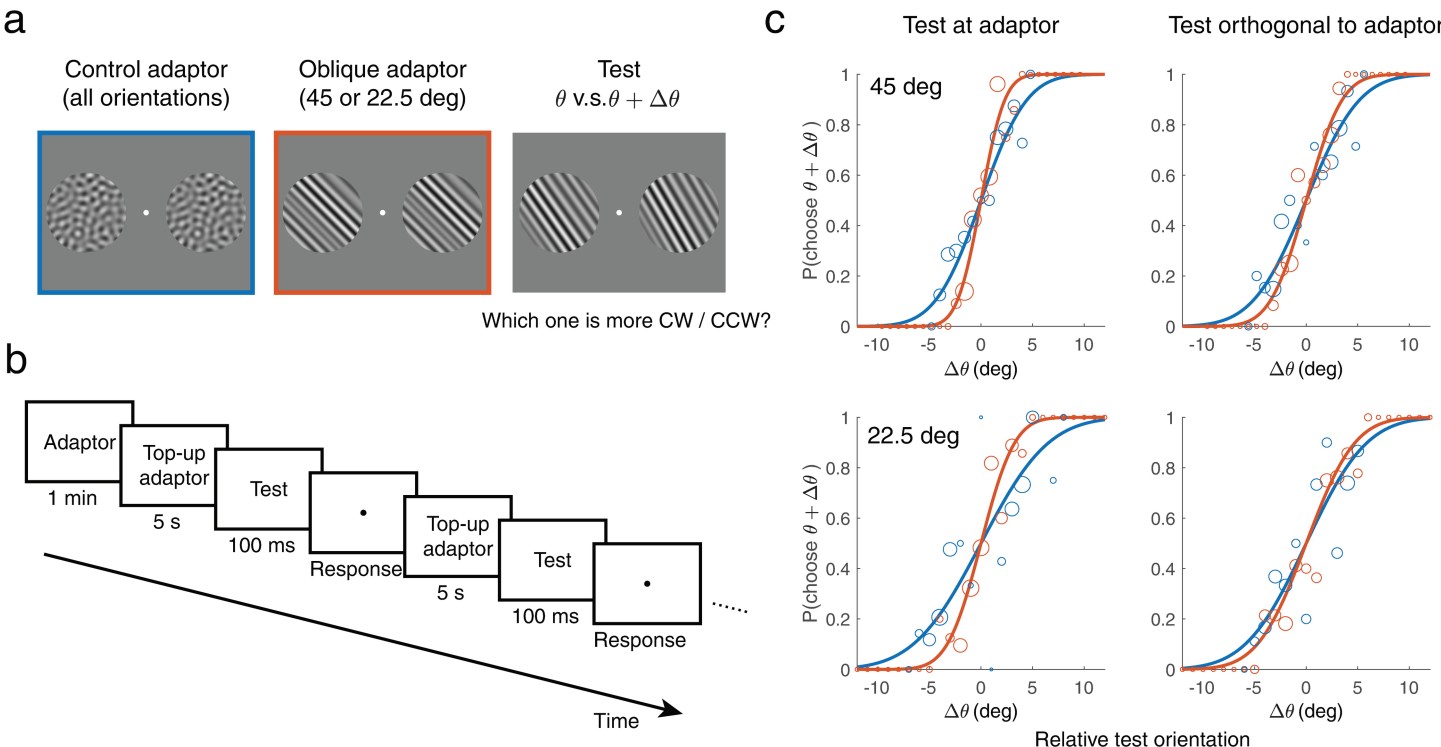

**Fig 2. Experimental procedure and measured adaptation-induced changes in discriminability.** (a) Stimuli were bandpass filtered white noise with either a uniform (control adaptor) or narrowband orientation spectrum (oblique adaptor, test stimuli). Two circular areas left and right of the fixation mark were simultaneously adapted before presentation of the two test stimuli of the 2AFC discrimination task at the same locations. (b) Task structure (single block). At the beginning of each block, there was an adaptation period of 1 minute. Every trial started with 5s top-up adaptation, after which subjects were briefly presented with two test stimuli and asked to report which one was more clockwise/counter-clockwise. Subjects performed the same number of trials for each of the four adaptor conditions (two oblique and their respective control adaptor conditions). (c) Example data and fitted psychometric curves at two test orientations of one subject (Subject 1). Adaptation to the oblique adaptors (orange) results in steeper psychometric curves for test orientations at and orthogonal to the adaptor orientation compared to the control adaptor condition (blue). Size of data points is proportional to the number of trials at that test orientation (adaptive staircase procedure).

adaptation. Each adaption condition (two oblique and their respective control adaptors) was measured over 8 blocks, resulting in 32 blocks corresponding to 6144 trials in total.

We tested two types of adaptors. The oblique adaptor was identical in structure to the test stimuli (i.e., same spatial bandpass and narrow orientation filter), with the orientation filter centered either at ±45 deg or ±22.5 deg (0 deg being vertical). We chose oblique as opposed to cardinal adaptor orientations to avoid possible ceiling effects; discrimination thresholds are known to be lowest at cardinal orientations [36] and adaptation is expected to lower them further at the adaptor orientation [6,7]. The second, control adaptor was identical to the oblique adaptor except that it had a uniform orientation spectrum. Discrimination thresholds measured in this well-defined, control adaptation condition serve as reference against which we compare changes in discrimination threshold induced by the oblique adaptors. Because the control adaptor is identical to the two oblique adaptors in every stimulus aspect except its orientation spectrum, it allows us to isolate the orientation specific, adaptation-induced changes in sensory encoding precision.

We independently fit psychometric curves to the 2AFC data (Fig 2c), extracted discrimination thresholds, and plotted discriminability as the inverse of the thresholds in order to have a direct comparison with encoding precision (Eq (1); Fig 3). In the control adaptation

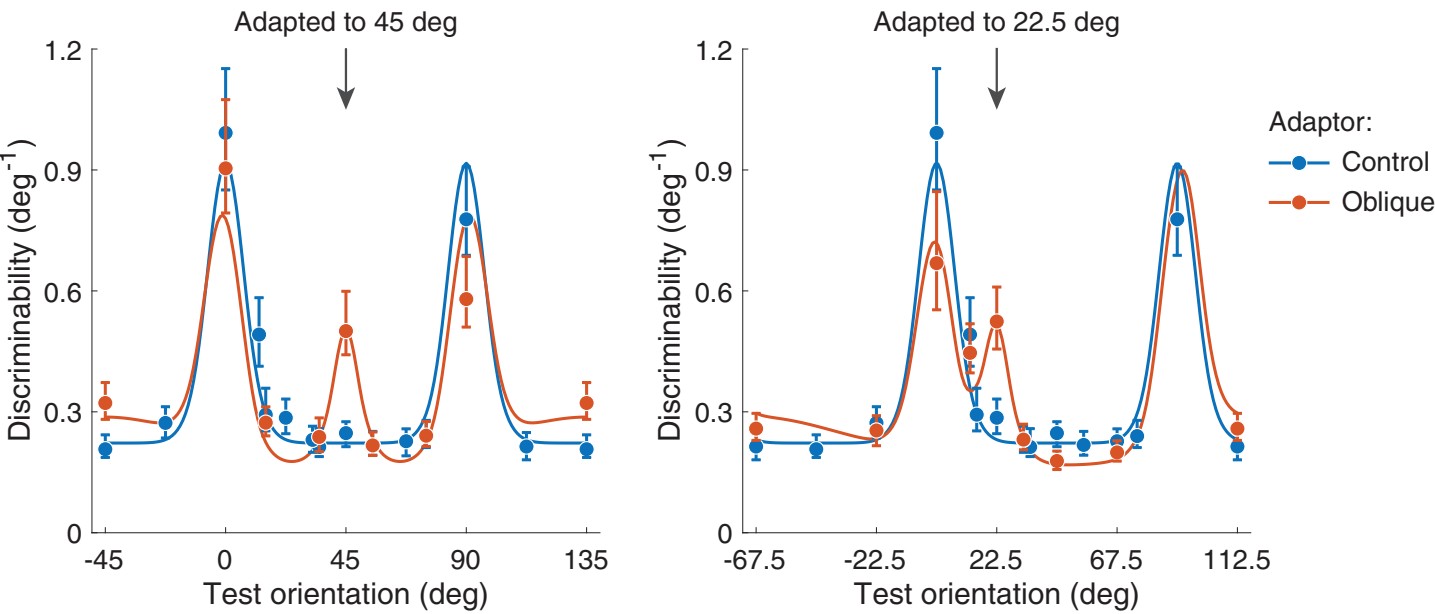

**Fig 3. Discrimination data and model fits averaged across subjects (see Fig 4 for individual data).** With the control adaptor (blue), discriminability is higher at cardinal orientations (0 deg is vertical). With the oblique adaptors (orange), discriminability increases at and orthogonal to the adaptor, and slightly decreases in between. Error bars represent the 95% intervals computed over 1000 bootstrap samples of the data (individual subjects).

condition, subjects' discriminability is highest at cardinal orientations as shown in previous studies without adaptation [36]. Adaptation to an oblique adaptor causes discriminability to increase at but decrease slightly away from the adaptor orientation compared to the control condition. This is consistent with previous findings [6,7]. Interestingly, discriminability for test orientations orthogonal to the adaptor also slightly increased after adaptation, which has not yet been consistently shown [2,7,33,34]. These general results hold for both oblique adaptors and across subjects (Fig 4).

## Model fit and comparison

As our experimental results show, adaptation improves discriminability for test orientations both at and orthogonal to the adaptor orientation. We thus assume an adaptation kernel that peaks at and orthogonal to the adaptor orientation (Fig 1). Specifically, we model the adaptation kernel as the weighted sum of two von Mises distributions and a uniform distribution. Widths and relative weights are free model parameters (see Methods). We individually fit the reallocation model to the data from each subject. We first fit the data measured under the control condition, which allowed us to determine the distribution of Fisher information before adaptation (i.e., every subject's individual sensory space $\tilde{\theta}$) and the overall amount of coding resource available (i.e., the subject's total Fisher information). We then jointly fit the data from the 45 deg and 22.5 deg adaptor conditions and determined the adaptation kernel of every subject. Fig 3 shows the average measured and predicted discriminability based on model fits to individual subjects' data. Fig 4 shows the measured and predicted discriminability of individual subjects. The model captures not only the improvement in discriminability at and orthogonal to the adaptor orientation, but also its mild deterioration for test orientations slightly different from the adaptor orientations. Note that for reasons of simplicity we assume encoding precision to be symmetric for vertical and horizontal orientations in the control

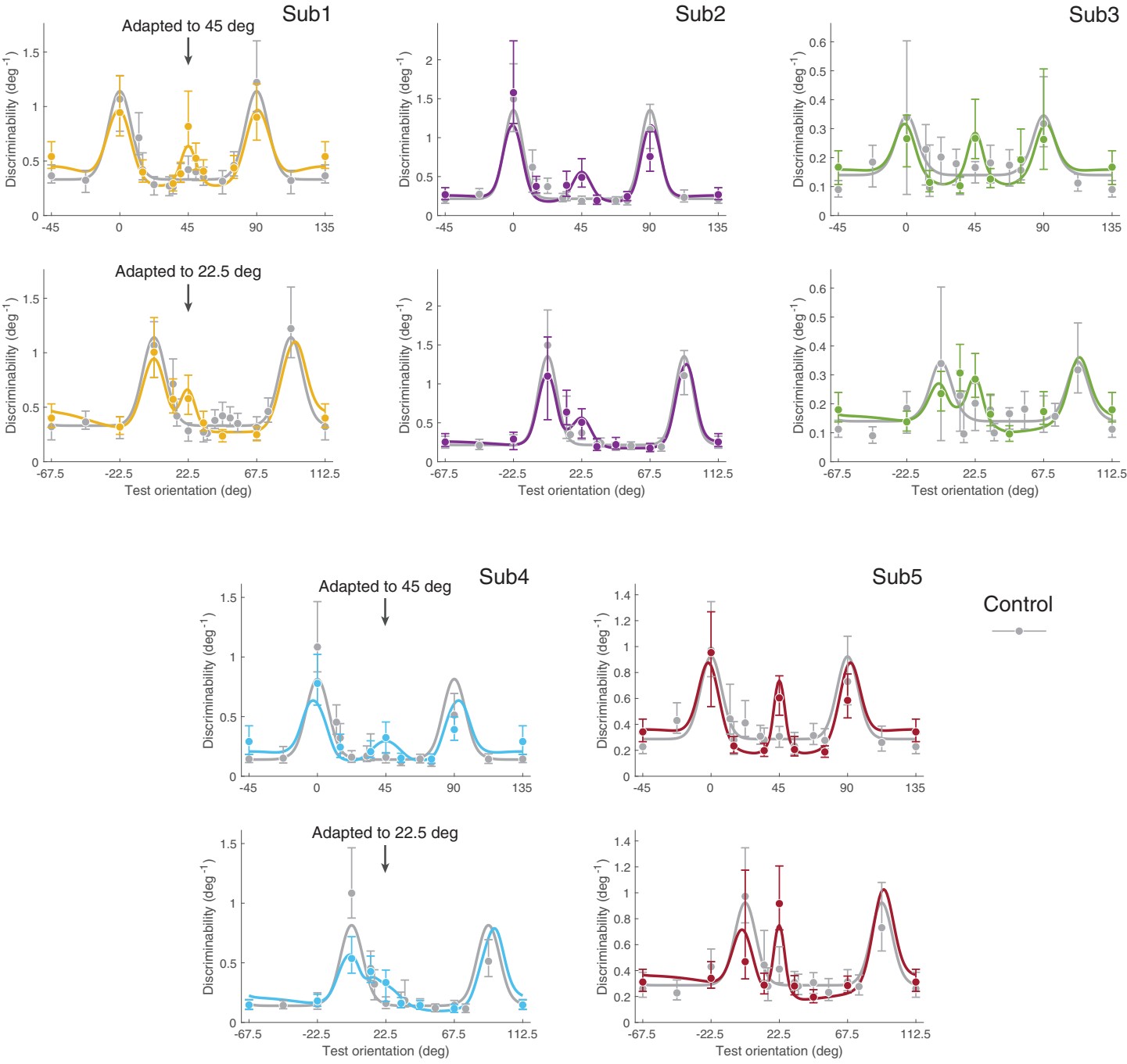

**Fig 4. Discrimination data and model fits for individual subjects.** Compared to the control condition (gray), subjects show higher discriminability at and orthogonal to the adaptor after adaptation, and lower discriminability slightly away from the adaptor (0 deg is vertical). Error bars represent the 95% intervals over 1000 bootstrap samples of the data. Table A in S1 Text: Best-fitting model parameters. Figs A and B in S1 Text: Individual psychometric functions (data and fits). Fig C in S1 Text: Discrimination threshold changes (adaptation vs. control condition).

adaptor condition, which does not fully agree with every subjects (e.g., subject 4). However, allowing for asymmetries in the stationary encoding model does not significantly change our results.

Fig 5a shows the adaptation kernels for every subject. The kernels were extracted from the fit reallocation model as illustrated in Fig 1. The kernels are similar across subjects, consistently showing a sharp peak in Fisher information at the adaptor as well as a more shallow peak for orientations orthogonal to the adaptor. Note that the kernels are plotted in each subject's individual sensory space $\tilde{\theta}$, determined by their encoding precision measured in the control adaptor condition.

We can further test key assumptions of the proposed reallocation model. For example, a separate fit to the data obtained in the two oblique adaptor conditions results in kernels that do not differ much from the kernels obtained from a joint fit, suggesting that adaptation is indeed governed by a mechanism that is independent of the specific adaptor orientation (Fig 5a). Also, when fit separately, the total Fisher information under oblique vs. control adaptor conditions is very similar, thus confirming our assumption that the total coding resource does not change with the adaptation state (Fig 5b). Finally, we performed a formal model test comparing the proposed reallocation model (2-peak) with a model that assumes coding improvement only at but not orthogonal to the adaptor (1-peak), a model that allows the total Fisher information to change with adaptation (2-peak + Fisher), and a model that allows the adaptation kernel to be different for different adaptor orientations (2-peak + kernel). When appropriately penalized for the number of free parameters (BIC), the reallocation model (2-peak) best fits the data for all subjects except Subject 2 for whom the only weak threshold improvement orthogonal to the adaptor favors an adaptation kernel with a single peak at the adaptor (Fig 5c).

## Natural scene statistics

Next, we analyzed the temporal statistics of visual orientations in the retinal image stream of freely behaving human observers. In particular, we extracted the orientation distribution in the next image frame after the human observers naturally experienced prolonged exposure to retinal inputs with a relatively static orientation content. The goal was to test whether these distributions match the changes in encoding precision (adaptation kernels) we extracted from the psychophysical adaptation data, and thus support the hypothesis that adaptation optimizes sensory encoding for statistically more likely, future stimuli.

Subjects took a stroll through a forest while wearing a head-mounted camera and an eye-tracker, simultaneously recording the scene and their eye movements, respectively (Fig 6a). In every frame of the video, we considered a central image patch (6x6 deg visual angle around the gaze center; Fig 6b) and extracted the visual orientation at every location within the image patch based on a linear multi-scale, multi-orientation image decomposition framework [58]. We computed orientation mean and variance over a sliding 3s time-window, and then identified instances where the variance at a particular location in the patch was small (circular variance smaller than 0.1). For those positions, we calculated the difference between the orientation in the next frame and the mean orientation over the preceding time-window (i.e., the adaptor orientation). Preferably, we would have considered a time-window that matched the adaptation duration in the psychophysical experiments (i.e., 60 seconds). However, under natural free-viewing conditions, it is very rare to observe instances where orientation in the retinal image stream is stable over such long durations. Therefore we chose a window-size that provides a balanced trade-off between providing sufficient data and being as long as possible.

Our analysis showed that at the same local position, the orientation in the next frame is most likely similar to the mean orientation in the immediate past (Fig 6c, left). The less stable the history (larger variance), the wider the orientation distribution in the next frame, until it eventually becomes uniform (Fig 6c, right; Fig D in S1 Text). Although a more detailed

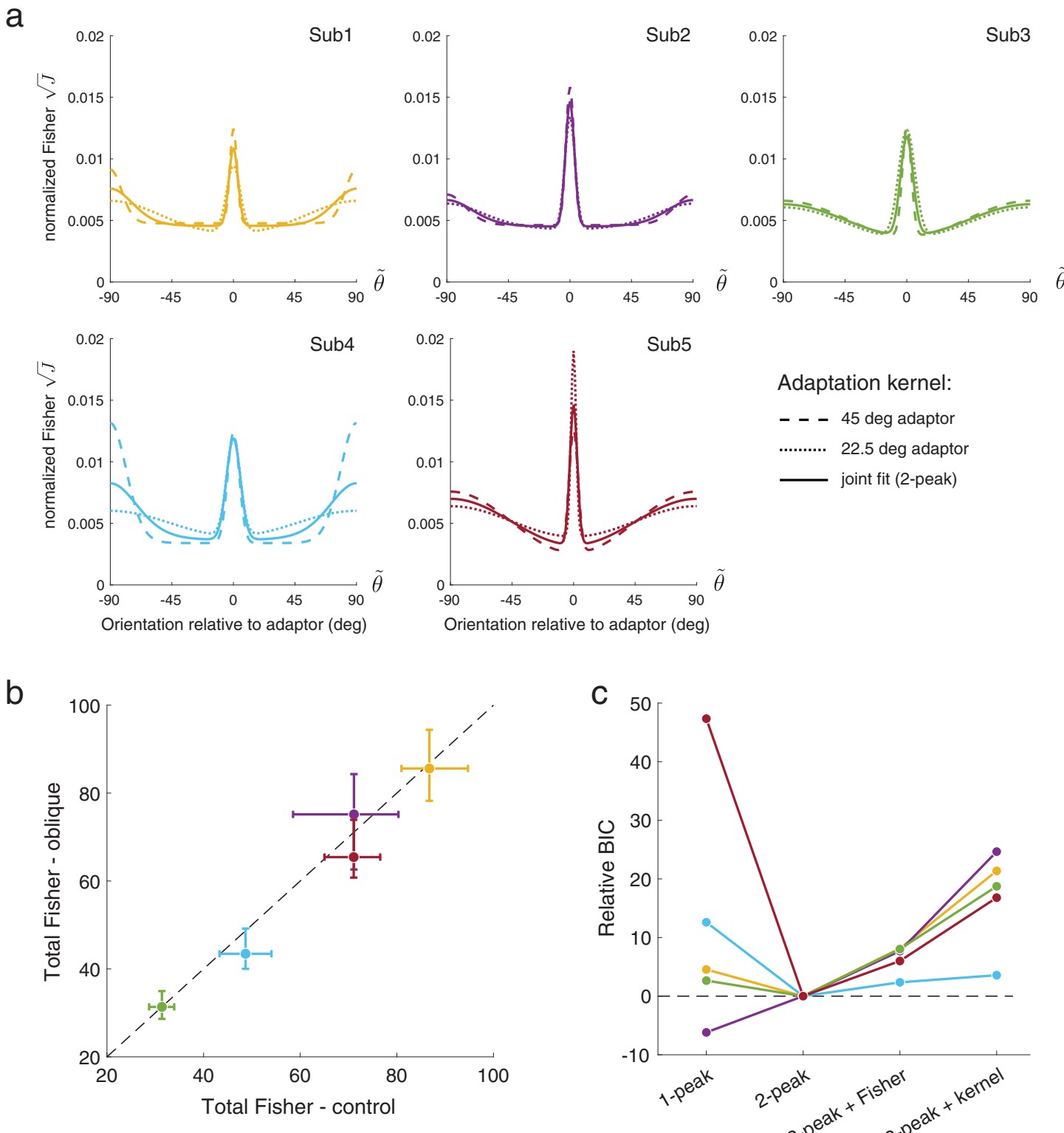

**Fig 5. Adaptation kernels.** (a) Extracted adaptation kernels, plotted in individual subjects' sensory spaces. Kernels are similar across subjects and do not substantially differ when separately fit to data from the two oblique adaptor conditions. (b) Total Fisher information (i.e., total coding capacity) extracted from separate fits to control and oblique adaptor conditions. Total values vary across subjects but fall close to the unity line. Error-bars represent 95% confidence intervals from 200 bootstrap samples of the data. (c) Testing the key assumptions of the reallocation model using a BIC goodness-of-fit comparison: original model (2-peak), relaxing the fixed resource assumption (2-peak + Fisher), relaxing the single kernel assumption (2-peak + kernel), and assuming a kernel with only a peak at the adaptor (1-peak). Subject color code as in Fig 4. Table B in S1 Text: Best-fitting model parameters (all model variants).

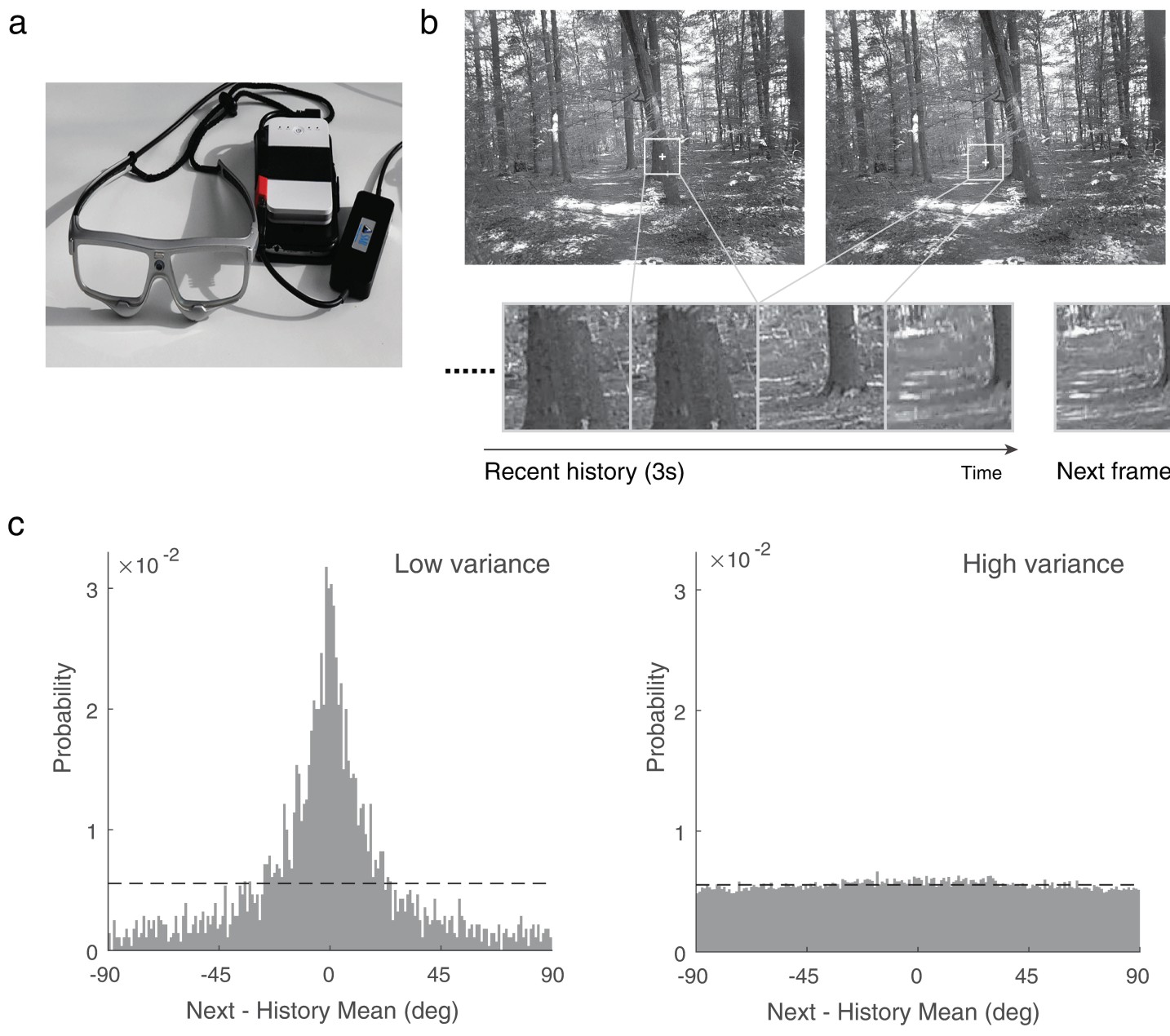

**Fig 6. Retinal input statistics under natural viewing condition.** (a) The combination of head-mounted camera and mobile eye-tracker allowed us to extract the retinal input statistics of human subjects freely behaving (i.e., walking) in a natural, forest environment. (b) At each local position within a small patch centered at the subjects' gaze location (white frame), we computed the distribution of local visual orientation in the next frame relative to the mean orientation over an immediately preceding 3s time-window. (c) Distributions of orientations in the next frame relative to the history mean when the variance of orientations in the time-window was low (<0.1; left) or high (>0.9; right), respectively. The two variance condition qualitatively correspond to the two adaptation conditions in our psychophysical experiment (oblique/control adaptor). Distributions represent the combined distributions across all spatial positions within the patch and across all spatial scales of the image decomposition (Methods). Fig D in S1 Text: Distributions for different history variances and spatial frequencies.

quantitative comparison is difficult due to the various differences in stimulus and adaptation conditions between free-viewing natural retinal input and our psychophysical experiments, the measured distributions suggest that reallocating Fisher information towards the adaptor orientation is consistent with optimizing encoding for future stimuli. For highly variable input

histories the distributions are mostly uniform and therefore a reallocation of resources is not useful given that the orientation in the next frame is essentially uncorrelated with past orientations; this corresponds to the control adaptor condition in our psychophysical experiment. Notably, the measured distributions do not show an increased probability for future stimuli at orientations orthogonal to the adaptor orientation (see Discussion).

## Predictive neural network

Previous studies have shown that deep neural networks implicitly learn to encode stimulus features as predicted by efficient coding [59,60]. Here we use "PredNet", a recurrent neural network designed and trained to predict the next frame in a video sequence [49,50], to test to what degree sensory representations dynamically change depending on the temporal input statistics. PredNet was inspired by the concept of "predictive coding" in neural networks [61]. PredNet uses top-down connections conveying the local predictions of incoming stimuli and bottom-up signals of the deviations from these predictions (Fig 7a). There are four sub-layers in each layer $i$ of PredNet: a recurrent representation layer ($R_i$), a prediction layer ($\hat{A}_i$), an input or target layer ($A_i$), and an error layer ($E_i$). The representation layer takes feedback from the error layer and the higher representation layer and generates predictions for the input layer; the error layer calculates the deviation of these predictions from the input and passes them on to the next input layer. Importantly, PredNet only contains static (i.e., non-adaptive) neurons. Previous studies have shown that PredNet reports illusory motions similar to those perceived by human subjects [62,63]. Here, we specifically investigate how the prolonged exposure to an adaptor stimulus affects the encoding precision in PredNet's representational layers.

We used PredNet pretrained on video sequences from the KITTI natural scenes data set [64]. We presented the network with the same stimulus sequence as used in our human adaptation experiment (Fig 7b), with the exception that the spatial frequency spectrum of the stimuli were matched to the average spectrum of the images in the training data set. We input a sequence of four adaptor frames (mimicking the adaptation phase) followed by a test frame, and then computed Fisher information in the lowest representation layer in response to the test stimulus. With the assumption that independent Gaussian noise corrupts the response in each neuron, Fisher information is equivalent to the squared gradient of the neural response in the direction of the test orientation. In this way, we computed Fisher information in the first representation layer as a function of the test orientation $\theta$ for all three adaptor conditions (control; 45 deg and 22.5 deg oblique adaptor).

Fig 8a shows PredNet's encoding precision when plotted in stimulus space. In the control adaptor condition, Fisher information in the first representational layer of PredNet is higher at cardinal compared to oblique orientations, albeit there is a noticeable asymmetry between horizontal and vertical encoding precision. After adaptation to the oblique adaptors, however, Fisher information peaks at the adaptor orientation. Both effects qualitatively match the measured changes in encoding accuracy in human subjects (Fig 4). Furthermore, when plotted in sensory space $\tilde{\theta}$ as defined by the networks' encoding precision measured in the control adaptor condition, Fisher information resembles the adaptation kernels we extracted from human subjects (Fig 5a). For both oblique adaptor conditions, the curves show a similar symmetric peak in encoding accuracy at the adaptor orientation and also the slight reduction in accuracy for orientations in the vicinity of the adaptor (Fig 8b).

The fact that PredNet, when exposed to similar input sequences, develops very similar changes in encoding accuracy to those we measured in human observers suggests that these encoding changes indeed help to better predict future sensory input of dynamic natural

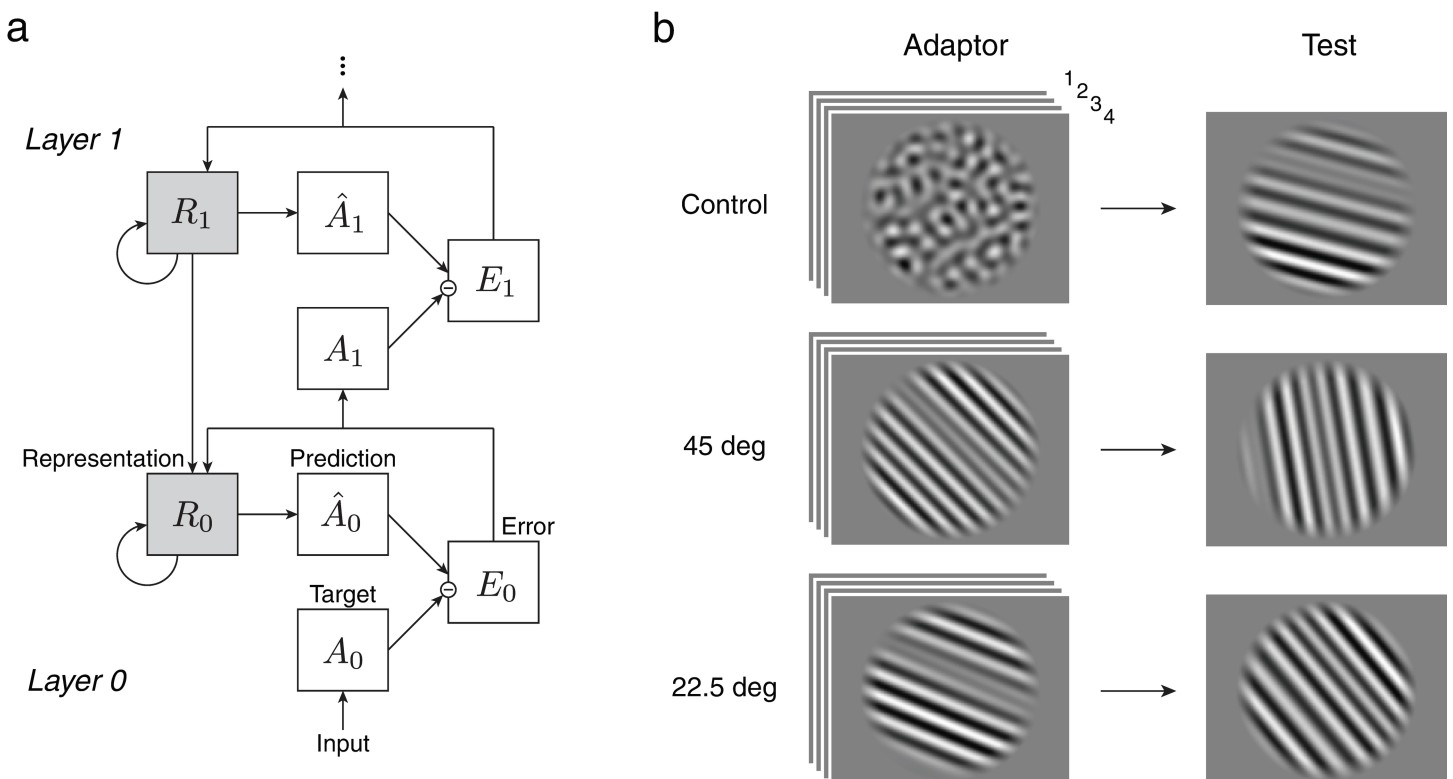

**Fig 7. Encoding changes in PredNet.** (a) Architecture of PredNet [49]. Each layer of PredNet consists of four sub-layers: a representation layer ($R_i$), a prediction layer ($\hat{A}_i$), an input or target layer ($A_i$), and an error layer ($E_i$). The representation layer takes feedback from the error layer and the higher representation layer and makes predictions about the next input. The error layer takes the difference between the prediction and the input and passes it on to the next layer. The network has four layers in total. (b) Adaptation experiment with PredNet. In each trial, PredNet is presented with four frames of the adaptor stimulus (control and oblique adaptors), followed by one frame of the test stimulus. The stimuli were the same as in the human psychophysical experiment (Fig 2). Encoding accuracy was measured based on the network's activity in the lowest representational layer in response to the test stimulus orientation.

scenes. Thus the adaptation induced reallocation of encoding capacity not only improves orientation encoding of more likely future stimuli, as shown with our natural image analysis, but also seems helpful to better predict the next visual stimulus.

## Discussion

In this paper, we provide converging evidence that adaptation in the human visual system adjusts sensory encoding accuracy such that it is optimized for future stimuli. We psychophysically measured adaptation-induced changes in sensory encoding accuracy of visual orientation in human subjects. We found that these changes are best described as a reallocation of sensory coding resources according to an isomorphic kernel that peaks at the adaptor orientation. Analyzing the temporal statistics of the retinal input of freely behaving human observers revealed that the distribution of local visual orientations in the next retinal input also depends on the immediately preceding stimulus history: the distribution shows a sharp peak at the mean orientation of a relatively stable stimulus history, but approaches a uniform distribution for histories with increasingly larger variances. These distributions qualitatively match the psychophysically measured changes in encoding accuracy, both for the control and the oblique adaptors. It suggests that adaption efficiently reallocates sensory coding

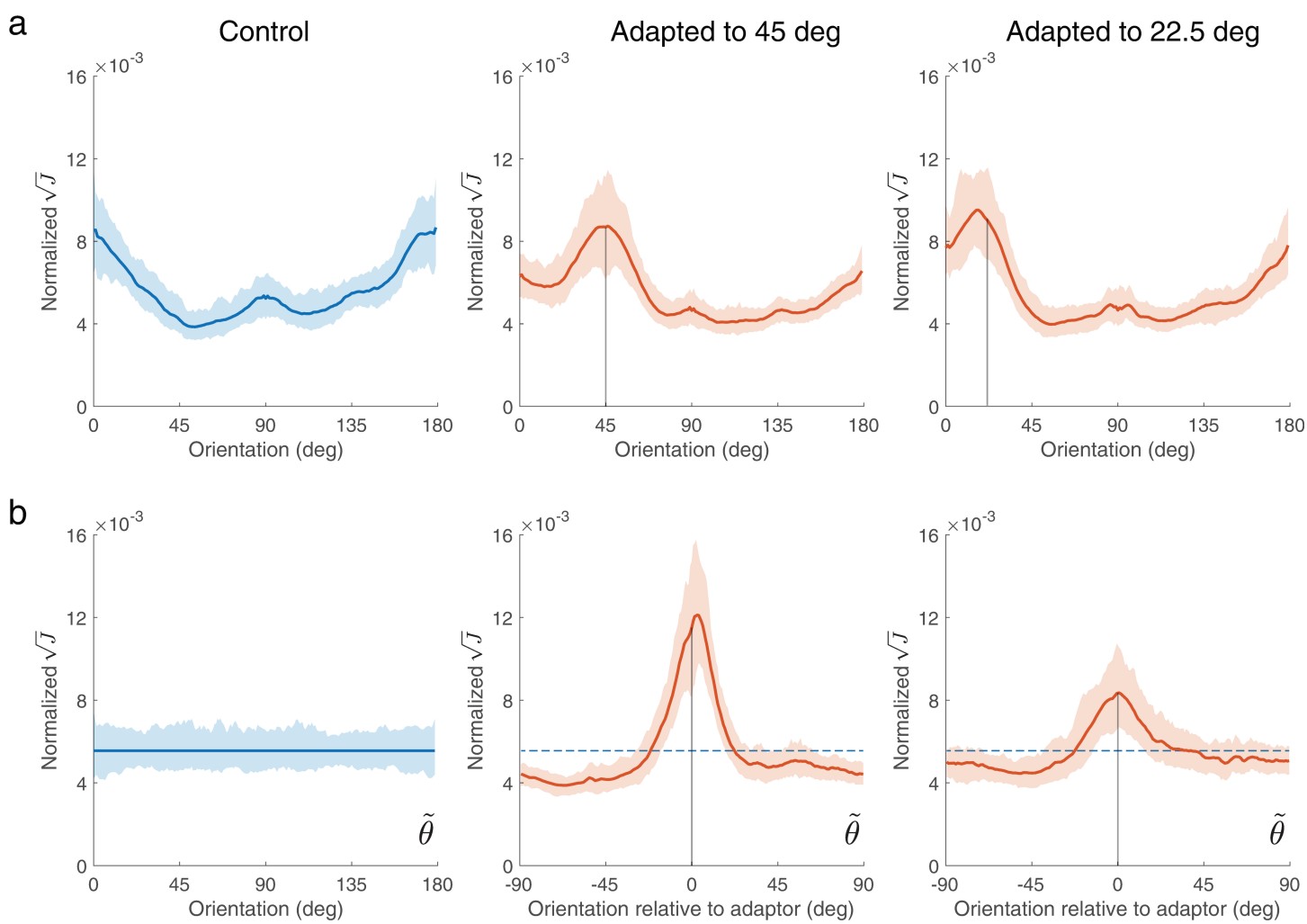

**Fig 8. Encoding accuracy in the first representational layer $R_0$ of PredNet after adaptation.** (a) Normalized square root of Fisher information as a function of test orientation (0 deg is vertical). In the control adaptor condition (blue curve) Fisher information is higher at cardinal orientations, reflecting the fact that PredNet creates efficient representations of visual orientation given the predominance of cardinal orientations in natural scenes. After adaptation to a 45 deg (middle) or 22.5 deg (right) oblique adaptor, however, Fisher information peaks at the adaptor orientation. (b) Same as in (a) but plotted in sensory space $\tilde{\theta}$. Sensory space is defined as the transformation $\tilde{\theta} = F(\theta)$ that leads to a uniform Fisher information distribution under the control adaptor condition (see also Fig 1). When plotted relative to the adaptor orientation, Fisher information in sensory space is qualitatively similar to the adaptation kernels of human subjects (see Fig 5a): Fisher information is peaked at and symmetric about the adaptor orientations, but reduced for test orientations further away. Lines and shaded areas represent the mean and the 95% confidence intervals over 200 stimulus sequences, respectively.

resources for future retinal input depending on the specific temporal structure of the preceding input history. Finally, we asked whether, and if so how, a recurrent neural network that is optimized to predict the next frame of natural scene videos changes its sensory encoding accuracy when being exposed to the same stimulus sequences used in our psychophysical adaptation experiments. We found encoding changes in the representational layers of the network similar to those we measured in human observers. Taken together, our results suggest that adaptation-induced changes in sensory encoding improve encoding accuracy for future stimuli, which in turn is beneficial for predicting future sensory input.

Although previous studies have measured changes in discriminability as a result of adaption to a single oriented stimulus [6,7], none of them did so across the full range of orientations and against a well-defined control adaptor condition. Interestingly, we found that adaptation not only increases discriminability at but also orthogonal to the adaptor orientation, thereby confirming previous evidence [2,7]. This rules out earlier, theoretically motivated proposals suggesting that the adaptation kernel should rather resemble a Difference-of-Gaussians (DOG) [51,52]. However, we currently lack a normative explanation of this orthogonal improvement because neither the retinal input statistics nor the encoding changes in PredNet show an effect at orthogonal orientations. We suspect that the orthogonal improvement is caused by the particular way visual orientation is encoded in neural populations, likely in combination with the specific mechanisms involved in increasing encoding precision at the adaptor orientation [65]. Future studies will be needed to fully uncover the origin of the orthogonal improvement.

Temporal statistics of visual orientation in video sequences of natural environments have been previously investigated, although not at the level of the retina. For example, [66,67] analyzed the recordings taken by a camera mounted on a cat's head. They found that at the same image location, prominent orientations are more likely to repeat themselves at short compared to long timescales. Another study analyzed video footage taken with a static camera in a natural environment [42]. It showed that dominant orientations in the next frame are more likely to be similar to the dominant orientations in the previous frame. Both of these results are qualitatively in agreement with our findings and are not unexpected given the mostly continuous and temporally smooth statistics of natural environments [43,65]. However, human vision is based on active sensing [44–46,48], where the active control of head- and eye-movements reshapes the statistical structure of the retinal input [38,39]. Thus, it is important to measure these statistical dependencies in the retinal input stream under natural behavioral and environmental conditions. A limitation of our current data set is that it mainly consists of forest scenes. Previous studies found that the overall orientation statistics in natural scenes somewhat depend on whether they contain man-made objects or not, as well as whether they are from indoor or outdoor environments [35]. While it seems unlikely that the temporally conditioned distributions we focus on are substantially different under different environmental contexts, future studies are necessary to evaluate this in more detail [68].

Artificial neural networks have become a useful model framework to test normative explanations of neuronal and behavioral phenomena [69,70]. The general rationale is that if a network, optimized for a certain functional goal within a certain training environment, shows some emergent encoding characteristics, then they are likely beneficial for the network in achieving said goal in said environment. Previous studies have shown that sensory representations in artificial neural networks trained to identify objects in natural scenes are similarly shaped by the statistical context (i.e., priors) of the stimulus environment as the representations in the human visual system [59,71]. Here we further demonstrate that this similarity extends to dynamical changes in these representations when the neural network is optimized for making predictions in a dynamic natural environment (PredNet). This supports the efficient coding hypothesis and proposes a functional role of sensory adaptation in making sensory prediction. Note that PredNet does not contain adaptive neurons. Rather, the changes in encoding emerge through the dynamic information flow in its recurrent architecture during the presentation of the adaptor stimuli. This also shows that the functional goal of sensory adaptation is independent of a specific neural adaptation mechanism; e.g., neuronal gain changes [4] may only be one of several ways to implement this goal.

The separation between function (i.e., encoding principle) and implementation (i.e., neural mechanisms) allows for a more general yet at the same time also more parsimonious definition of adaptation. Adaptation has been frequently characterized in terms of how it changes the tuning characteristics of neurons. In addition to a reduction in neural gain, multiple effects such as changes and shifts in tuning curves [1,4] as well as the homeostatic control of the neural populations activity have been attributed to adaptation [72]. However, these effects seem specific to certain cortical areas [5] and probably also certain animal species [73], and thus likely depend on the specific biophysical constraints and limitations. Here we show that adaptation can be understood on the basis of a single computational principle that, however, may rely on different specific neural implementations case by case. In that regard our results complement previous work on modeling adaptation as a trade-off between neural encoding costs and information loss [74,75]. At the level of implementations these trade-offs may play an important role (see e.g., orthogonal coding improvement). Our results provide new insights about the functional objectives that guide such trade-off considerations.

It is worth further discussing the idea of the sensory space for which we characterize the adaptation kernel (Fig 1). The space reflects the efficient sensory representation of the stimulus feature given the overall, stationary stimulus statistics in the environment [23,28]. The representational geometry of the space is such that under stationary conditions the statistics and the encoding precision are homogeneous and uniform. We propose to think of adaptation as a transient modulation or fine-tuning of this geometry such that it is optimally suited for the next stimulus given the shortterm stimulus context. Note also that the adaptation kernel is only isomorphic when adapting to very narrow shortterm distributions (i.e., approximate a Dirac function). More broadly distributed adaptor orientations (stimulus space), however, will result in shortterm distributions in sensory space that are warped according to the transformation $F(\theta)$, and thus are generally different depending on where they are located in orientation space. We hypothesize that the adaptation kernels we derived from our experiments represent a unitary descriptions of localized orientation adaptation. Adaptation induced encoding changes for arbitrary adaptor distributions can then be determined as the convolution of those shortterm distributions (in sensory space) with the unitary adaptation kernel. Future work is necessary to test this hypothesis.

Finally, we can extend our view of adaptation to more generally construed, statistical contexts. For example, it is well documented that spatial context can change sensory encoding of visual orientation in ways similar to temporal context, both at the neural and behavioral level [65,76]. The changes are also well-aligned with the fact that in natural scenes, visual orientation at an image location is best predicted by the average orientation within its surround [41]. Furthermore, these spatial contexts are specific to the particular image content, and the associated modulation of neural response patterns can be explained with a flexible gating mechanisms that is optimally tuned for these specific contexts [77]. Thus we conjecture that temporal adaptation is but one mechanism with which neural systems ensure that their representation of sensory information is efficient with regard to the statistical context of their environment.

**Conclusions**  Efficient coding has been a prominent hypothesis for sensory adaptation. The present study measured the changes in coding accuracy induced by the prolonged exposure to a static adaptor stimulus, and found a universal parametric description of the reallocation of coding resources under such conditions. Analysis of the temporal statistical structure of the retinal image stream in freely behaving humans and measurements of sensory representations in recurrent neural networks provide converging support for the efficient coding

hypothesis of adaptation: adaptation-induced changes in encoding accuracy reflect the visual systems' attempt to best possibly represent the next expected sensory input.

## Methods

### Ethics statement

All experiments were approved by the Institutional Review Board of the University of Pennsylvania under protocol # 850568. All human subjects provided written informed consent. No AI tools have been used for any aspect of this research and its presentation.

### Psychophysical experiment

**Subjects.** 5 subjects (2 female), 25 to 33 years old, participated in the experiment. Subject 1 was non-naive. All subjects had normal or corrected-to-normal vision. They were remunerated at a rate of 10$ per hour, plus a 20$ bonus upon completion of the full experiment.

**Setup.** The experiment was run using Matlab (R2016b) with the PsychToolbox [78]. Subjects sat in a darkened room and viewed stimuli on a VPixx3D screen (1920×1080 pixels resolution, 120 Hz refresh rate) at a 89 cm distance. A circular aperture (26 cm diameter) was placed in front of the screen to occlude the edges of the screen removing any potential cardinal orientation cues.

**Stimuli.** All stimuli were presented on a gray background with mean luminance 40 cd/m$^2$. Stimuli consisted of filtered white noise patterns with same overall mean luminance as the background. For all stimuli (control and oblique adaptors, test), the noise was first filtered with a band-pass filter with uniform power spectrum across the spatial frequency range of 3.75–5.25 cpd. Oblique adaptors and test stimuli were then further filtered with an oriented filter with a symmetrically warped Laplace profile centered at the desired orientation and a standard deviation of 1.4 deg. Stimuli had 80% contrast. Stimuli were 2 deg in diameter, presented 1.67 deg to the left and right of fixation. A fixation dot was presented at the center of the screen throughout the experiment.

**Procedure.** The experiment was organized in blocks. At the beginning of a block, subjects viewed one of the adaptor stimuli for 60 s. After this initial adaptation phase, each trial started with top-up adaptation (5 s), followed by a blank interval (0.35 s), presentation of the test screen (0.1 s), and a response period. During the initial adaptation phase and the top-up period, two identical adaptor patterns were presented on both sides of fixation, refreshing every 1.25 s with a 0.05 s blank frame in between. In the test screen, a test and a reference stimulus were presented to the left and right of fixation, randomly assigned. During the response period, subjects pressed one of two buttons on a gamepad to indicate which stimulus in the test screen was more clockwise (or counterclockwise, interleaved across blocks). Subjects did not receive feedback after the response. Subjects were instructed to maintain fixation throughout the initial adaptation phase and during each trial. Our choice of initial adaptation duration and top-up adaptation is fairly typical and follows previous adaptation studies (see e.g., [7]).

The experiment consisted of two parts, each of which contained an oblique adaptor condition and a control adaptor condition. In the first half of the experiment, the oblique adaptor was oriented at ±45 deg (vertical corresponds to 0 deg). The test orientations were [0, ±10, ±30, ±45, 90] deg relative to the adaptor orientation (subject 1 had ±5 deg in addition). In the second half of the experiment, the oblique adaptor was oriented at ±22.5 deg. The test orientations were [0, ±10, ±22.5, ±45, 90] deg relative to the adaptor orientation. Test orientations were randomized across trials. The reference orientation varied according to a 2-up-1-down

staircase procedure in 25 equal steps within a $\pm9.6$ deg, $\pm15$ deg, $\pm18$ deg, or $\pm24$ deg range relative to the test orientation, depending on the performance of each subject in the training session prior to the experiment.

Subjects completed 192 trials for each test orientation in each adaptation condition (216 trials for subject 1 in the first half of the experiment). In each half of the experiment, subjects completed the control adaptor condition first followed by the oblique adaptor condition. In the oblique adaptor condition, subjects completed 4 blocks of one adaptor orientation, then 4 blocks of the same adaptor orientation but mirrored around vertical (6 blocks each for subject 1 in the first half). The order of the two adaptor orientations were counterbalanced across subjects. The control adaptor condition consisted of 8 blocks (12 blocks for subject 1 in the first half). Each block lasted for about 25 min. Blocks with different adaptors (control versus oblique adaptor condition, or opposite oblique adaptor orientations) were completed at least one day apart.

## Data analysis

For the main analysis, data in blocks with -45 or -22.5 deg oblique adaptors were combined with data from blocks with 45 or 22.5 deg adaptor by mirroring it across vertical. Psychometric curves were obtained by fitting cumulative Gaussian distributions to the data. We assumed a zero-mean Gaussian distribution and no lapse rate. Discrimination thresholds were calculated at the 75% level based on the fitted psychometric functions.

## Modeling

**Pre-adaptation (control adaptor).** Let $\theta$ be the orientation of the test stimulus and $m$ its sensory measurement in a given trial. Before adaptation (i.e., defined by the control adaptor condition) the discrimination threshold is typically lower at cardinal orientations [36], which implies higher Fisher information at cardinal orientations [23]. We parametrized the square root of the Fisher information distribution $J(\theta)$ as the weighted sum of a uniform distribution and two identical von Mises distributions centered at two cardinal orientations:

$$\sqrt{J(\theta)} \propto k\,\mathrm{vm}(\theta; 0, \kappa) + k\,\mathrm{vm}(\theta; \pi, \kappa) + \frac{1 - 2k}{2\pi}, \tag{3}$$

where $\kappa$ represents the width of the distribution around cardinal orientations, and $k$ represents the relative amplitude. Note that because angles are defined on $[-\pi, \pi]$, 0 and $\pm\pi$ represent cardinal orientations (0 corresponds to vertical). We use this convention throughout the paper.

We consider encoding under the control condition as reflected by a sensory space (stationary) in which Fisher information is uniform. The mapping $\tilde{\theta} = F(\theta)$ from stimulus to this sensory space is the cumulative of the square root of Fisher information distribution [28], thus $F(\theta) \propto \int_{-\pi}^{\theta} \sqrt{J(\chi)}\mathrm{d}\chi$. Assuming that sensory noise in this space follows a von Mises distribution, we can write the measurement distribution in sensory space as

$$p(\tilde{m}|\tilde{\theta}) = \mathrm{vm}(\tilde{m}; \tilde{\theta}, \kappa_i), \tag{4}$$

where $\kappa_i$ represents the sensory noise magnitude. The distribution in stimulus space $p(m|\theta)$ directly follows by transformation according to the inverse mapping $\theta = F^{-1}(\tilde{\theta})$.

**Post-adaptation (oblique adaptors).** After adapting to a single orientation $\theta_a$, the distribution of the square root of Fisher information in sensory space changes accordingly to

$$\sqrt{J_a(\tilde{\theta}; \theta_a)} \propto p_a(\tilde{\theta} - \tilde{\theta}_a), \tag{5}$$

where $\tilde{\theta}_a = F(\theta_a)$ is the adaptor orientation in sensory space, and $p_a$ is the adaptation kernel. Thus, adaptation reallocates Fisher information by the same adaptation kernel shifted according to the adaptor. Now, the stationary sensory space is not a uniform space any more. However, we can apply another transformation $\theta^* = F_a(\tilde{\theta}) \propto \int_{-\pi}^{\tilde{\theta}} \sqrt{J_a(\chi)} d\chi$ to obtain an adapted sensory space. Importantly, we assume that in the transformed space sensory noise remains von Mises distributed with the same internal noise parameter as in the stationary sensory space (Eq 4), thus the total coding resource does not change after adaptation (reallocation). We can write the measurement distribution as

$$p(m^* | \theta^*) = \text{vm}(m^*; \theta^*, \kappa_i), \tag{6}$$

where $\kappa_i$ represents the constant sensory noise magnitude. The distribution in stimulus space $p(m|\theta)$ then follows by successive transformations according to the inverse mappings $\tilde{\theta} = F_a^{-1}(\theta^*)$ and $\theta = F^{-1}(\tilde{\theta})$.

The "2-peak" (original) model assumes that the adaptation kernel $p_a(\tilde{\theta})$ is a weighted sum of two independent von Mises distributions and a uniform distribution:

$$p_a(\tilde{\theta}) = k_1 \, \text{vm}(\tilde{\theta}; 0, \kappa_1) + k_2 \, \text{vm}(\tilde{\theta}; \pi, \kappa_2) + \frac{1 - k_1 - k_2}{2\pi}, \tag{7}$$

where $k_1$ and $k_2$ represent the relative amplitudes, and $\kappa_1$ and $\kappa_2$ the widths of the two peaks.

The "1-peak" model assumes that $p_a(\tilde{\theta})$ is the weighted sum of one von Mises distribution and a uniform distribution:

$$p_a(\tilde{\theta}) = k_1 \, \text{vm}(\tilde{\theta}; 0, \kappa_1) + \frac{1 - k_1}{2\pi}, \tag{8}$$

where $k_1$ and $\kappa_1$ represents the relative amplitude and the width of the peak, respectively.

The "2-peak + Fisher" model assumes that total Fisher information can change after adaptation, i.e., the sensory noise (i.e. the width of the von Mises likelihood) in the adapted sensory space $\kappa_i^a$ is allowed to be different from the the noise in the stationary-adaptation sensory space $\kappa_i$.

The "2-peak + kernel" model permits that adaptation kernels $p_a(\tilde{\theta})$ for the 45 deg and 22.5 deg adaptor condition can be different.

**Discrimination decision and response distribution.** Let $\theta_t$ and $\theta_r$ be the orientation of the test and reference stimulus, and $m_t$ and $m_r$ their sensory measurements respectively. The probability of the reference orientation being clockwise of the test orientation can be calculated as

$$p(\theta_t - \pi < \theta_r < \theta_t | m_t, m_r) = \int_0^{2\pi} p(\theta_t | m_t) \int_{\theta_t - \pi}^{\theta_t} p(\theta_r | m_r) \, d\theta_r \, d\theta_t. \tag{9}$$

If the probability is larger than 0.5, the observer would make the decision that the reference orientation is clockwise (cw), otherwise counter-clockwise (ccw) of the test orientation.

Because test and reference stimuli were always tested for the same adaptation state of the observer, had the same probability to be more clockwise or counter-clockwise, and the stimuli had the same spatial frequency, contrast and presentation duration, the decision process can be simplified to a direct comparison of the measurements of the two stimuli: if $m_r$ is clockwise of $m_t$, the subjects would make the decision that the reference orientation is clockwise of the test orientation, and vice versa. So the decision probability of the reference orientation being clockwise of the test orientation can be written in terms of the measurement distributions (Eq (4) and (6) transformed to stimulus space), thus

$$p(\text{``}\theta_r \text{ is } CW\text{''}|\theta_t, \theta_r) = p(m_t - \pi < m_r < m_t|\theta_t, \theta_r) = \int_0^{2\pi} p(m_t|\theta_t) \int_{m_t-\pi}^{m_t} p(m_r|\theta_r) \, \mathrm{d}m_r \, \mathrm{d}m_t.$$

(10)

## Model fitting

We fit the model by finding the model parameters $\rho$ that maximize the likelihood of the model given the data $D$:

$$p(D|\rho) = \prod_{j=1}^{n} p(D^j|\rho) = \prod_{j=1}^{n} p(r^j|\rho, \theta_t^j, \theta_r^j),$$

(11)

where $\theta_t^j$ and $\theta_r^j$ are the test and reference orientations, $r^j$ is the response in trial $j$, and $n$ is the total number of trials.

We first fit the model to the control adaptor condition with the following free parameters:

- $\kappa_i$ for sensory noise;
- $k$ for the relative peak amplitude, and
- $\kappa$ for the width of the Fisher information distribution (cardinal orientations).

Then we fix these parameters and fit the model to the oblique adaptor condition.

The 2-peak model has four free parameters for the adaptation kernel:

- $k_1$ and $k_2$ for the relative amplitudes of the two peaks;
- $\kappa_1$ and $\kappa_2$ for the width of the two von Mises distribution.

The 2-peak + Fisher model has an additional parameter $\kappa_i^a$ allowing a different sensory noise after adapting to an oblique adaptor (i.e. change in overall coding resource). The 2-peak + kernel model has four parameters for the adaptation kernel of each oblique adaptor, eight in total. The 1-peak model has only two free parameters for the strength and width of the adaptation kernel.

## Natural scene statistics

**Data set.** The dataset was obtained with a video-based, portable eye tracker system (Eye Tracking Glasses 2 from SensoMotoric Instruments). Videos were recorded by the head-mounted camera while subjects were freely walking in a forest environment. Image sequences had a resolution of 1280x960 pixels at 24 frames per second, spanning 60x46 deg of visual angle. The compression algorithm was H.264. Synchronized eye movements were binocularly measured at 120 samples per second. Initial calibration of each subject was accomplished with a 3 point calibration target. Subjects subsequently wore the eye tracker for at least 10 minutes

before checking the calibration again. The resulting eye tracking accuracy was well below 1 degree at about 10m distance. We included videos from 9 subjects, with a total length of 12 minutes. Videos were converted to grayscale.

**Data analysis.** We looked at a 6x6 deg area centered at the gaze location in each frame of the videos. We extracted the orientation at each position within the area using a steerable pyramid image decomposition [58]. The steerable pyramid provides a decomposition on the basis of k-th order orientation filters at different spatial scales. We rotated and applied 1st-order steerable filters to find the orientation with the strongest response as the orientation of each position. We then computed the orientation mean and circular variance over a sliding 3s time-window (72 frames) at each position, and computed the difference between the orientation in the next frame and the mean orientation in the previous 3s. We performed these computations independently at each position in levels 2 to 4 of the steerable pyramid; each level operates on an image half the resolution of the one used in the preceding level. This corresponds to 64x64, 32x32, and 16x16 positions, respectively. In Fig 6, we included time and positions where the history variance is smaller than 0.1 or larger than 0.9, respectively, and combined the data from three levels.

## PredNet

PredNet is a recurrent neural network that is trained to predict the next frame of a video. We used PredNet pretrained on the KITTI data set for our experiment [49,50,64].

**Stimuli.** The stimuli were images of filtered white noise patterns with a size of 128x160 pixels. The control adaptors were filtered by the spatial frequency spectrum extracted from the original training data set within the range of 8-12 cycles per image. The spatial frequency filter was obtained by taking the average of the 2D Fourier transformation of the images across all frames and averaging across orientation for each spatial frequency, with a low and high cutoff at 8 and 12 cycles per image, respectively. The oblique adaptors and test stimuli were further filtered by an orientation filter with a symmetrically wrapped Laplace spectrum centered at the desired orientation with a standard deviation of 1.4 deg; this is the same orientation filter as used for the psychophysical adaptation stimuli (see above). Stimuli had 100% contrast. The noise pattern was embedded in a circular aperture at the center of the image; the contrast of the noise pattern faded linearly from 100% to 0 as the distance from the center increases from 48 to 60 pixels.

**Procedure.** Each input sequence consisted of four adaptor frames followed by a test frame. Four adaptor frames were sufficient to elicit a substantial adaptation effect; using 10 frames did not significantly change the results. We fed this five-frame input sequence to Pred-Net and extracted the activation of the first representational layer in response to the test frame (frame #5). For each adaptation condition (control, oblique 22.5 and 45 deg), we tested 200 input sequences (different noise patterns, exact same filter properties). To compute Fisher information as a function of test orientation, we rotated the test frame in each sequence in 1 deg intervals.

**Calculating Fisher information.** We computed Fisher information in the first representation layer ($R_0$) as a function of orientation $\theta$ in the test frame (Fig 7a). Assuming independent Gaussian noise, Fisher information can be calculated for each test frame as

$$J(\theta) = \sum_i \left( \frac{\partial r_i}{\partial \theta} \right)^2 \tag{12}$$

where $r_i$ is the response of the $i$th unit in layer $R_0$. Because we focus on the distribution of coding resources, we normalized the sum of the square root of Fisher information across orientation. Fig 8 shows the mean and 95% confidence intervals over the 200 input sequences.

## Supporting information

**S1 Text. Supplementary Tables A and B; Supplementary Figs A–D.** Individual subjects' data and fitting parameters; detailed natural scene statistics.
(PDF)

## Acknowledgments

The authors thank the members of the Computational Perception and Cognition Laboratory for many fruitful discussions and constructive feedback.

## Author contributions

**Conceptualization:** Jiang Mao, Alan Alfred Stocker.

**Data curation:** Jiang Mao, Constantin A. Rothkopf.

**Formal analysis:** Jiang Mao, Alan Alfred Stocker.

**Funding acquisition:** Alan Alfred Stocker.

**Investigation:** Alan Alfred Stocker.

**Methodology:** Jiang Mao, Constantin A. Rothkopf, Alan Alfred Stocker.

**Project administration:** Alan Alfred Stocker.

**Resources:** Constantin A. Rothkopf, Alan Alfred Stocker.

**Software:** Jiang Mao.

**Supervision:** Alan Alfred Stocker.

**Validation:** Jiang Mao, Alan Alfred Stocker.

**Visualization:** Jiang Mao, Alan Alfred Stocker.

**Writing – original draft:** Jiang Mao, Alan Alfred Stocker.

**Writing – review & editing:** Jiang Mao, Constantin A. Rothkopf, Alan Alfred Stocker.

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
