## [Decision Letter · Decision Letter 0]

2 Jul 2024

Dear Dr. Stocker,

Thank you very much for submitting your manuscript "Adaptation optimizes sensory encoding of future stimuli" for consideration at PLOS Computational Biology.

As with all papers reviewed by the journal, your manuscript was reviewed by members of the editorial board and by several independent reviewers. In light of the reviews (below this email), we would like to invite the resubmission of a significantly-revised version that takes into account the reviewers' comments.

We cannot make any decision about publication until we have seen the revised manuscript and your response to the reviewers' comments. Your revised manuscript is also likely to be sent to reviewers for further evaluation.

Sincerely,

Roland W. Fleming, PhD

Academic Editor

PLOS Computational Biology

Lyle Graham

Section Editor

PLOS Computational Biology

Reviewer's Responses to Questions

**Comments to the Authors:**

Reviewer #1: The paper argues that perceptual adaptation in humans is optimized to accurately represent expected, future stimuli. This work is a very nice mix of different approaches: theoretical results, human psychophysics, analysis of natural stimulus statistics, and machine learning models of perception. Overall, I find these results interesting and appreciate the variety of techniques and methods used by the authors.

My biggest concern is that it is not immediately clear how different parts of the manuscript relate to its title, i.e., the predictive nature of perceptual adaptation, and how they relate to each other. I also think that the text is not clear enough and that the authors should significantly improve the presentation to facilitate the understanding of these results. Below I present some more specific concerns and suggestions:

1. The theory does not explicitly focus on the temporal dynamics of sensory stimuli. Please explain and clarify where the titular “future stimuli” are in the formalism. How do adaptation kernels and resource reallocation relate to prediction? In the current presentation, it is difficult to understand how the theory accounts for the dynamic nature of the stimuli. Where does the anticipatory/predictive component come from? I would welcome a better, more explicit explanation of these aspects.

2. The first part of the results (theory) is written in a very hermetic language that is not aimed at an interdisciplinary audience. For example, it is not at all clear what the difference between the “sensory space” and the “stimulus space” is. The authors refer a lot to the literature (including their previous work), but this cannot replace a good explanation of their theory. I suggest rewriting this section and giving more space to explaining the goal of the theory (e.g. in the current explanation, Fisher information suddenly changes to maximizing mutual information – line 128). I also suggest adding a diagram to Figure 1 explaining what stimuli are represented, what a “sensory space” is, etc.

3. Perceptual experiments are well captured by the theory, but again: how do they relate to temporal prediction? It is not explained with the necessary clarity. The reader is left to guess.

4. The section on natural scene statistics feels almost incomplete. Since a large part of it is devoted to describing the mechanics of computing orientations in natural video, it reads like a part of methods rather than results – the actual result is only two sentences (lines 225-230). The terminology in Figure 6 (e.g., “high history variance” vs. “low history variance”) is very specific to the image analysis method rather than results. What is also missing is an explicit connection – either mathematical or even graphical – to the theory presented in the first part. How do these statistics relate to theory, psychophysics, and prediction? There is a statement about adaptation kernels, but it is very verbal, short and unclear. I believe this section should be significantly improved.

5. The same criticism applies to the description of PredNet. I appreciate the Fisher information analysis of the network. However, it is presented almost as an independent fact. In my opinion, it should be clearly explained how the goal of PredNet (reconstructing the next frame of a movie) is related to the theory presented (Fisher information maximization) and the perceptual results. I don't think this is done sufficiently well at the moment.

Reviewer #2: The authors argue that the goal of sensory adaptation (to orientation) is to allocate information processing resources to likely upcoming visual input. They use psychophysical data and visual scene statistics in the presence of eye movements to support their argument. In addition, they demonstrate that a recurrent artificial neural network designed to predict the next input also shows a similar pattern of adaptation.

The strength of the study is that concepts that have been linked informally in the past, are linked explicitly, with a solid computational framework, and strong supporting experimental data.

My comments are mainly requests for clarification and discussion:

The measured thresholds and fits support the main hypothesis. However, thresholds were not always collected at all relevant test orientations (and this seems to have varied across subjects and even adapters -see “missing” thresholds at some of the troughs of the fitted curves). Please explain why. Obviously this is not ideal as it implies that the trough is driven solely by model assumptions of symmetry). This is further complicated by the observation that most subjects show asymmetries in the thresholds that are not well captured by the hypothesis (e.g., higher thresholds on one “side” of the adapter than the other. I’d recommend doing a (model-free) analysis of the thresholds at the single subject level to document (a)symmetry and potentially identify other aspects of the data that are statistically reliably but not capture by the normative model. This does not distract from the overall value of the normative model, but provides nuance to the claim that adaptation is only about optimizing for the future input.

The description of the histories is not quite clear. A sentence like L279-L281seems tautological in isolation. Pleas explain how time (or space?) windows were selected to generate the low/high variance conditions. Why are the distributions for different spatial frequencies relevant for the case made in this manuscript? A sentence or two on the nature of the different levels of the steerable pyramid would also help the reader.

L305. Is the contrast of the control adapter really identical for a v1 receptive field? To me this seems unlikely as the combination of orientations tends to cancel out some of the peaks and troughs. Please quantify.

Figure8 – it may be helpful to draw a thin blue line (representing the control condition) in the adaptation panels of b (it took me a while to realize that the orthogonal directions were indeed reduced relative to control).

Figure5 - 100 bootstrap samples - that seems an unusually small number for a confidence interval.

What motivated the choice to use four “adaptation” frames in PredNet? Is there some way to relate this to time? Does the effect change with the number of frames?

Reviewer #3: PLoSCompBio

This work examines the phenomenon of adaptation to visual sensory stimuli. The authors measure behavioral adaptation in psychophysics experiments involving subtle discrimination of oriented bars. Participants show sharper psychometric curves around the target (adapted) orientation but also around the orthogonal orientation. The authors finally use a predictive coding model to capture the behavioral effects. There also an interesting measurement and discussion of natural image statistics and dynamic changes through head mounted displays that include eye tracking. This part is quite interesting but not well integrated with the rest of the paper (not even mentioned in the abstract, except in a highly indirect manner). Overall, the results are interesting and provide insights about the mechanisms underlying adaptation in sensory processing.

1. If I understand Fig. 1 correctly, the sensory enhancement (middle row) at the adapted orientation is narrow whereas the corresponding enhancement at the orthogonal orientation is rather broad (according to the reallocation model).

a. The caption to Fig. 1 seems to suggest that Fisher information (J(theta)) is shown, but the y-axis in the figures says Fisher sqrt(J). In any case, there is no real quantitative y-axis, so the figures could be reflecting J or sqrt(J) or any other function.

b. It would be nice to explain why is this the case

c. Is it possible to test this prediction experimentally as well?

d. The y-axis in Fig. 1 does not have any numbers so I am not sure how seriously one should take the idea beyond the conceptual level. In addition to the broadening, the peak seems to be larger at the target orientation compared to the orthogonal one. To the extent that the magnitude of this peak is relevant, one may deduce that there would be more modulation at the target orientation than at the orthogonal one. But this prediction does not seem to be supported by the results in Fig. 2.

2. The test in Fig. 2 involves some parameter choices. While I am perfectly ok with those choices, I wonder how the authors decided them. Did they try different parameters and converged on a set of parameters that work. Would the results of adaptation be stronger with longer adapting times? The authors discuss eye tracking in the context of other experiments but do not seem to be mention whether subjects maintained fixation during the entire 1 min of adaptor time + 5s top-up adaptor.

3. I wonder whether models that incorporate adaptation would reproduce the experimental results in Fig. 2/3. For example, Vinken et al 2020 (Incorporating intrinsic suppression in deep neural networks captures dynamics of adaptation in neurophysiology and perception) developed a model that incorporated adaptation into a visual recognition neural network. I guess that this type of model would reproduce the effects at target orientation but I am not sure about the orthogonal one. Similar questions apply to other adaptation models.

4. The authors have not made data or code available. They indicate that they will upon acceptance. In my opinion, thisi not ideal practice and code/data should be submitted for review as well.

**Have the authors made all data and (if applicable) computational code underlying the findings in their manuscript fully available?**

Reviewer #1: Yes

Reviewer #2: **No:** I did not see an explicit reference to a data/analysis sharing location

Reviewer #3: **No:** I made a comment about this in the comments to author.s.

PLOS authors have the option to publish the peer review history of their article (what does this mean?). If published, this will include your full peer review and any attached files.

Reviewer #1: No

Reviewer #2: No

Reviewer #3: No
---

## [Decision Letter · Decision Letter 1]

21 Dec 2024

Dear Dr. Stocker,

We are pleased to inform you that your manuscript 'Adaptation optimizes sensory encoding for future stimuli' has been provisionally accepted for publication in PLOS Computational Biology.

We also suggest that you consider the final remarks of Reviewer 1 as you prepare the manuscript.

Best regards,

Lyle Graham

Section Editor

PLOS Computational Biology

Reviewer's Responses to Questions

**Comments to the Authors:**

Reviewer #1: I would like to thank the authors for taking my suggestions into consideration. I believe that the clarity of the manuscript has substantially improved. As I mentioned in my first review - this paper nicely combines computational and perceptual work and I'm certain it will be of interest to a broader community. I therefore support publication of the manuscript.

I have three minor comments that the authors may want to take into consideration prior to publication:

1) The assumption of the model is that the adaptor effectively changes the expected distribution of stimuli to a mixture of the marginal and the adaptor. An alternative is that it would be *only* a distribution around the adaptor (without the marginal). Perhaps this would be worth commenting on?

2) I think that the prediction aspect is perhaps overaccentuated - after all even the analysis of natural scene statistics indicates that orientations eithed remain practically constant in the regime consistent with the predictions. The temporal prediction here is of the trivial kind - just expect the same stimulus. At the same time - it is not inconsistent with the data and the authors are fully entitled to this interpretation.

3) Fig 6c - perhaps it would make sense to visualize the conditional distribution of orientations for both contexts (i.e. p(orientation_{t+1} | mean(history))? I would find it both interesting and helpful.

**Have the authors made all data and (if applicable) computational code underlying the findings in their manuscript fully available?**

Reviewer #1: Yes

PLOS authors have the option to publish the peer review history of their article (what does this mean?). If published, this will include your full peer review and any attached files.

Reviewer #1: No

---

## [Editor Report · Acceptance letter]

PCOMPBIOL-D-24-00755R1

Adaptation optimizes sensory encoding for future stimuli

Dear Dr Stocker,

I am pleased to inform you that your manuscript has been formally accepted for publication in PLOS Computational Biology. Your manuscript is now with our production department and you will be notified of the publication date in due course.

With kind regards,

Olena Szabo
